# Effect of Spaceflight on Tomato Seed Quality and Biochemical Characteristics of Mature Plants

**Elena Dzhos [1], Nadezhda Golubkina [1],\*, Marina Antoshkina [1], Irina Kondratyeva [1], Andrew Koshevarov [1], Anton Shkaplerov [2], Tatiana Zavarykina [3,4], Galina Nechitailo [3] and Gianluca Caruso [5]**

[1] Federal Scientific Center of Vegetable Production, Selectsionnaya 14, 143072 Moscow, Russia; elenadzhos@mail.ru (E.D.); limont_m@mail.ru (M.A.); tomatvniissok@mail.ru (I.K.); zato@inbox.ru (A.K.)
[2] Yuri Gagarin Cosmonaut Training Center, Star City, 141160 Moscow, Russia; shkaplerov@mail.ru
[3] Emanuel Institute of Biochemical Physics RAS, Kosigina 4, 119334 Moscow, Russia; tpalievskaya@yandex.ru (T.Z.); spacemal@mail.ru (G.N.)
[4] Federal Scientific Center "Nemchinovka", Agrokhimikov 6, Novoivanovskoe, 143026 Moscow, Russia
[5] Department of Agricultural Sciences, University of Naples Federico II, Portici, 80055 Naples, Italy; gcaruso@unina.it
\* Correspondence: segolubkina45@gmail.com

**Abstract:** Intensive space exploration includes profound investigations on the effect of weightlessness and cosmic radiation on plant growth and development. Tomato seeds are often used in such experiments though up to date the results have given rather vague information about biochemical changes in mature plants grown from seeds subjected to spaceflight. The effect of half a year of storage in the International Space Station (ISS) on tomato seeds (cultivar Podmoskovny ranny) was studied by analyzing the biochemical characteristics and mineral content of mature plants grown from these seeds both in greenhouse and field conditions. A significant increase was recorded in ascorbic acid, polyphenol and carotenoid contents, and total antioxidant activity (AOA), with higher changes in the field conditions compared to greenhouse. Contrary to control plants, the ones derived from space-stored seeds demonstrated a significant decrease in root AOA. The latter plants also showed a higher yield, but lower content of fruit dry matter, sugars, total dissolved solids and organic acids. The fruits of plants derived from space-stored seeds demonstrated decreased levels of Fe, Cu and taste index. The described results reflect the existence of oxidative stress in mature tomato plants as a long-term consequence of the effect of spaceflight on seed quality, whereas the higher yield may be attributed to genetic modifications.

**Keywords:** space-stored seeds; *Solanum lycopersicum* L.; weightlessness; cosmic radiation; antioxidants

## 1. Introduction

International space stations provide unique conditions for the investigation of the effects of space radiation and microgravity on plant growth and development [1–5]. Such experiments can serve as the basis for subsequent cultivation of vegetable crops in space during long-distance spaceflights, for understanding the mechanisms of physiological changes in plants and for evaluating the prospects of quick plant selection from space seeds [1,6–8].

Previous investigations of the effects of space radiation and microgravity on plant growth and development achieved on Russian and American space stations revealed insignificant changes in plant morphology, which suggests good prospects of plant cultivation in space [9,10]. Another direction of investigations includes studies of the effect of space on the quality of both the seeds and the plants grown from these seeds on the Earth. According to Liu et al. [11], this direction opens high prospects of quick plant selection aimed to increase yield, tolerance to diseases and vegetation period shortening. A series of such investigations were completed on different agricultural crops, and a beneficial

effect of spaceflight was demonstrated on rice seeds [11]. On the other hand, the results obtained in recent times are often controversial. Indeed, investigations on rocket seeds [12] showed that spaceflight reduced seed germination vigor and increased aging sensitivity but did not compromise seed viability and the development of normal seedlings. An investigation on tomato seeds after 6 years of spaceflight indicated that the tested plants exhibited higher variability in yield than the control ones, and some of the tested plants were infertile; moreover, various differences in cell walls, chloroplasts and mitochondria were observed. The results obtained point out significant changes occurring at the molecular level in tomato plants [10,13]. Experiments carried out in Ukraine [14] demonstrated that spaceflight conditions during a 6-year exposure of tomato seeds increased tomato productivity, whereas the plants were more resistant to viruses and had higher polyphenol concentration than those of the stationary control. In the early experiment of Kahn and Stoffella [15], the authors showed that tomato seeds could survive in space for several years without adverse effects on germination, emergence and fruit yield. Investigations of the effect of 15-day spaceflight revealed acceleration of alfalfa seed germination and inhibition of the root growth due to chromosomal damage and abnormal mitosis induced by cosmic radiation [16]. Other results revealed reduced germination, lethality, sterility and accelerated senescence [17–19].

The chronic exposure to low doses of ionizing radiation also led to significant differences in the expression of radical scavenging enzymes and DNA-repair genes and an increase in the activity of several antioxidant enzymes [20].

Long-term exposure to microgravity inside spaceships also resulted in the important discovery that these conditions are associated with accelerated aging of humans and plants [4,21]. The conditions of space stations reduced seed vigor and viability, which are connected with oxidation of the most important molecules: proteins, lipids and nucleic acids [22,23]. The best shielding for crop seed transport during long-distance space travel was the seed storage inside spaceships [12].

The aim of the present study was to conduct a quality evaluation of mature tomato plants grown from seeds exposed to half a year of spaceflight on the International Space Station (ISS).

## 2. Materials and Methods

Tomato seeds (*Solanum lycopersicum* L., cultivar Podmoskovny ranny of dwarf type) were obtained in 1992 and exposed in the cosmic station Mir (from 1992 to 1998). After returning to the Earth, 8 generations of plants were grown both in greenhouse and open field [13], and only the seeds of the 8th tomato generation were used in the present research. Seeds were transported to the International Space Station (ISS) on 19 December 2017 by the cosmonaut Shraplerov through the transportable manned spacecraft MS-07. Space-treated seeds were stored for six months inside the ISS, at the average temperature of 22–23 °C. The seed samples were brought back to the Earth on 3 June 2018. Control seeds of cultivar Podmoskovny ranny had been kept at room temperature in the laboratory of the Emanuel Institute of Biochemical Physics since 1998.

### 2.1. Growing Conditions and Experimental Protocol

Seeds of tomato (cultivar Podmoskovny ranny) kept for half a year in the ISS and seeds of control plants were used in the present investigation.

A research was carried out in 2020 on plants grown in (a) unheated film-covered greenhouse and (b) in open field, at the experimental fields of Federal Scientific Center of Vegetable Production (Moscow region, 55°39.51′ N, 37°12.23′ E), on sod-podzolic clay-loam soil, pH 6.8, 2.1% organic matter, 1.1 g·kg$^{-1}$ N, 0.045 g·kg$^{-1}$ P$_2$O$_5$, 0.357 g·kg$^{-1}$ K$_2$O. The mean values of temperature (°C) and relative humidity (%) were the following: 16.1 and 71.8 in May, 21.0 and 73.0 in June, 23.8 and 74.9 in July, 19.0 and 76.9 in August and 14.8 and 86.0 in September.

Upon the 22 April sowing, the control seeds showed 33% germination and space-treated seeds reached 61%. The seedlings were transferred into cassettes (5 × 5 cv) with a peat mix substrate containing mineral fertilizers and pH 6.5–7. Most of the seedlings emerged on 4 May and were transplanted in greenhouse on 28 May and in open field on 16 June, when they had 6–7 true leaves, with 3 plants per m², and each treatment was replicated thrice. Fertilization was practiced by supplying 30 kg·ha$^{-1}$ N (ammonium sulfate), 60 kg·ha$^{-1}$ $P_2O_5$ (superphosphate) and 100 kg·ha$^{-1}$ (potassium sulfate) prior to planting and 50 kg·ha$^{-1}$ N (ammonium nitrate) during the crop cycle in two applications, two and five weeks after transplant, respectively.

At harvest, started in early August, the following determinations were performed in all plots: plant height; weight of tomato fruits; number of marketable trusses per plant.

## 2.2. Resistance to Phytophtorosis

The evaluation of plant resistance to phytophtorosis was carried out during the natural development of the disease against a severe infection background, according to the guidelines for tomato selection relevant to phytophtorosis resistance [24].

## 2.3. Sample Preparation

Ten samples of healthy red-ripe stage fruit were used, harvested in August. Each sample consisted of at least three tomatoes from the second to fourth trusses, with a minimum total weight for sample of 250 g.

After harvesting, leaves, fruits and roots were separated and weighed; roots were washed with water and dried with filter paper. Samples were homogenized, and fresh homogenates were used for the determination of ascorbic acid, nitrates and total dissolved solids (TDS). Some of the samples were dried at 70 °C to constant weight and used for the determination of total polyphenol content (TP), total antioxidant activity (AOA) and mineral composition.

## 2.4. Dry Matter

The dry matter was assessed gravimetrically by drying the samples in an oven at 70 °C until constant weight.

## 2.5. Ascorbic Acid

The ascorbic acid content was determined by visual titration of plant extracts in 6% trichloracetic acid with Tillman's reagent [25]. Three grams of fresh tomato fruits were homogenized in a porcelain mortar with 5 mL of 6% trichloracetic acid and quantitatively transferred to a measuring cylinder. The volume was brought to 60 mL using trichloracetic acid, and the mixture was filtered through filter paper 15 min later. The concentration of ascorbic acid was determined from the amount of Tillman's reagent that went into titration of the sample.

## 2.6. Preparation of Ethanolic Extracts

Half a gram of dry homogenized tomato fruit or root powder was extracted with 20 mL of 70% ethanol at 80 °C over 1 h. The mixture was cooled and quantitatively transferred to a volumetric flask, and the volume was adjusted to 25 mL. The mixture was filtered through filter paper and used further for the determination of polyphenols and total antioxidant activity.

## 2.7. Total Polyphenols (TP)

Polyphenols were determined spectrophotometrically based on the Folin–Ciocalteu colorimetric method according to Golubkina et al. [26]. The concentration of polyphenols was calculated according to the absorption of the reaction mixture at 730 nm using 0.02% gallic acid as an external standard. The results were expressed in mg of gallic acid equivalent per g of dry weight (mg GAE g$^{-1}$ d.w.).

### 2.8. Antioxidant Activity (AOA)

The antioxidant activity was evaluated via titration of 0.01 N $KMnO_4$ solution with ethanolic extracts of dry samples [26].

### 2.9. Total Dissolved Solids (TDS)

TDS were determined in water extracts using TDS-3 conductometer (HM Digital, Inc., Seoul, Korea) and expressed in $mg \, kg^{-1}$ d.w.

### 2.10. Nitrates

Nitrates were assessed using ion-selective electrode on ionomer Expert-001 (Econix Inc., Moscow, Russia).

### 2.11. Monosaccharides (SS)

The monosaccharides were determined using the ferricyanide colorimetric method based on the reaction of monosaccharides with potassium ferricyanide [27]. The total sugars were analogically determined after acidic hydrolysis of water extracts with 20% hydrochloric acid. Fructose was used as an external standard.

### 2.12. Titratable Acidity (TA)

TA was determined potentiometrically by titrating a 50 mL diluted (1:5) sample with 0.1 N NaOH to pH 8.1 on ionomer Expert 001 (Econix Inc., Russia) and was expressed as percentage of citric acid.

### 2.13. Taste Index (TI)

TI was determined according to Navez et al. [28] from the total sugar content (TS) and TA values using the formula

$$TI = TA + TS/(20 \times TA). \tag{1}$$

### 2.14. Carotenoid Content

Determination of carotenoid content was achieved according to Golubkina et al. [26]. First, 0.5 g of homogenized sample was ground in a mortar with ceramic powder and extracted with small portions of acetone until color disappearance. The combined extract was diluted with 9 mL of hexane and washed 4–5 times with distilled water to remove traces of acetone. The residual extract was quantitatively transferred to a volumetric flask, and the volume was adjusted to 10 mL. The resulting extract was mixed, filtered through a small portion of anhydrous $Na_2SO_4$ and subjected to the analysis. The separation of carotenoids was achieved using quantitative thin-layer chromatography on Whatman 3A chromatographic paper in two chromatographic systems: (1) hexane to separate β-carotene and (2) hexane–acetone, 10:0.5, for separation of lycopene and lutein. Appropriate zones of carotenoid compounds were cut out and filled with 3 mL of hexane. The determination of carotenoid content in tomato fruit was performed using appropriate specific absorption $E^{1\%}_{1cm}$ for β-carotene (2580 at λ = 450 nm), lycopene (3470 at λ = 474 nm) and lutein (2560; λ = 447 nm). The internal standards were β-carotene, lutein and lycopene from Sigma Inc. (Kawasaki, Japan).

### 2.15. Statistical Analysis

Data were processed by analysis of variance, and mean separations were performed through the Duncan multiple range test, with reference to 0.05 probability level, using SPSS software version 21 (IBM, Armonk, NY, USA). Data expressed as percentages were subjected to angular transformation before processing.



## 3. Results and Discussion

Tomato plants grown both from control and space-stored seeds had a shorter crop cycle in greenhouse (90–94 days) compared to open field (115–120 days) (Table 1), due to the higher temperatures recorded in the first environment. The differences in crop cycle length between plants from space-stored and control seeds were 5 and 4 days in open field and greenhouse respectively.

**Table 1.** Phenological progress of tomato plants grown in greenhouse and in open field, from control and spaceflight-exposed seeds, expressed as days from sowing referring to 50% of plants that reached each stage.

|  | Control Seeds | | Space-Stored Seeds | |
| --- | --- | --- | --- | --- |
|  | **Greenhouse** | **Field** | **Greenhouse** | **Field** |
| Two-leaf stage | 13 | 13 | 12 | 12 |
| Flowering | 52 | 57 | 49 | 56 |
| Fruit ripening | 94 | 115 | 90 | 120 |

### 3.1. Yield, Dry Matter Content, TDS and Nitrates

Higher height, fruit yield and marketability were recorded for tomato plants grown from space-stored seeds compared to the control ones, with higher values in the more favorable growth conditions in greenhouse (Table 2). The plants derived from space-stored seeds also showed better tolerance to diseases in the field conditions compared to control plants, which was in accordance with the results of the previous investigation [14].

**Table 2.** Plant height, fruit yield, dry weight and disease occurrence of tomato grown in greenhouse and in open field, from control and spaceflight-exposed seeds.

|  | Control Seeds | | Space-Stored Seeds | |
| --- | --- | --- | --- | --- |
|  | **Greenhouse** | **Field** | **Greenhouse** | **Field** |
| Plant height (cm) | 65 b | 57 c | 77 a | 78 a |
| Yield (t ha$^{-1}$) | 39.0 b | 34.3 c | 42.0 a | 36.9 b |
| Fruit weight (g) | 58 b | 52 c | 63 a | 56 b |
| Number of trusses per plant | 5 a | 4 b | 5 a | 4 b |
| Marketability (%) | 89 ab | 85 c | 92 a | 87 bc |
| Diseases (number of points) | 3.0 a | 2.5 b | 3.0 a | 3.0 a |
| Dry matter (%) | 11.5 ± 1.0 a | 7.1 ± 0.6 bc | 8.2 ± 0.6 b | 6.4 ± 0.5 c |

In each row, the values with the same letters do not differ statistically according to Duncan test at $p < 0.05$.

At the same time, a significant decrease in fruit dry matter content was found in the fruits obtained from plants derived from space-stored seeds compared to control ones. Notably, in greenhouse conditions, the fruit dry matter associated with the seed space treatment was 1.4 times lower than that of control plant fruits, while the corresponding difference in the open-field-grown plants was only 1.12-fold.

At the same time, the total dissolved solids (TDS) did not differ statistically between fruits produced from control and those obtained from space-treated seeds, though the latter merely showed a slight decreasing tendency. TDS values detected by portable conductometer reflect both the amount of soluble solids and the organic acid content. As can be seen in Figure 1, higher TDS values were recorded in greenhouse conditions, which is connected with a higher nutrient uptake rate.

A similar phenomenon was observed for nitrate accumulation, though with greater differences between the control fruits and those produced by the plants derived from space-stored seeds (Figure 2). The latter results are not surprising, taking into account that all nitrate derivatives are highly soluble in water.

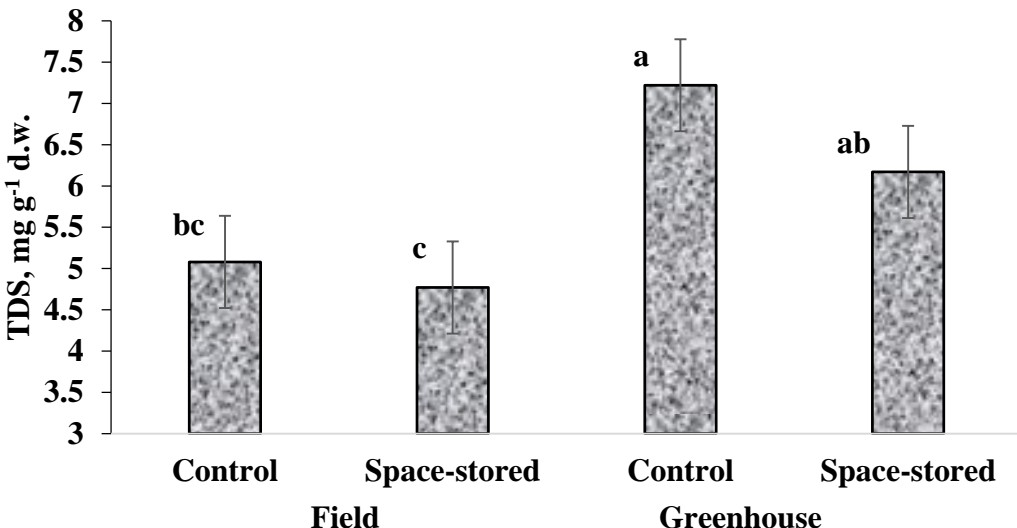

**Figure 1.** TDS content in tomato fruits grown from control and spaceflight-exposed seeds, in open field and in greenhouse. Values with the same letters do not differ statistically according to Duncan test at $p < 0.05$.

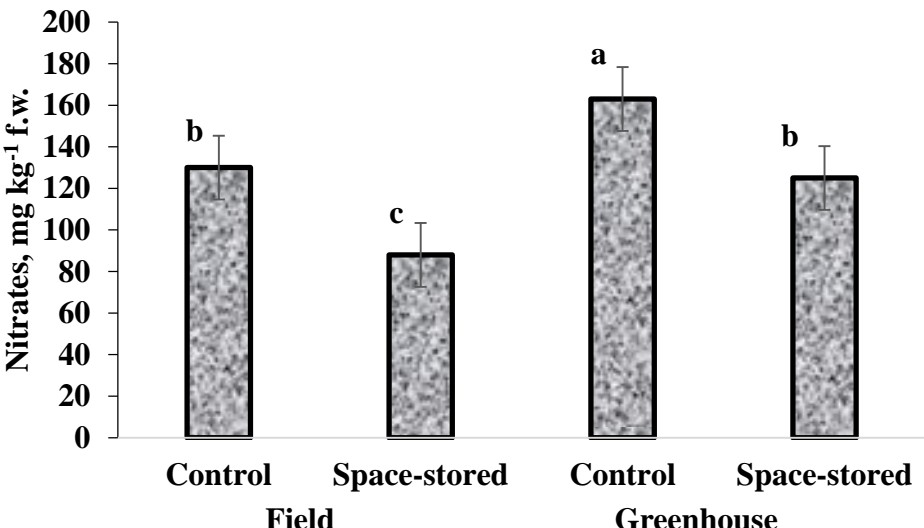

**Figure 2.** Nitrate accumulation in tomato fruits grown from control and spaceflight-exposed seeds, in open field and in greenhouse. Values with the same letters do not differ statistically according to Duncan test at $p < 0.05$.

### 3.2. Antioxidant Status

Secondary metabolites and antioxidants in particular are known to be involved in the process of stress adaptation [29]. Tomato fruits are rich in antioxidants, including carotenoids, polyphenols and ascorbic acid, demonstrating high biological activity, thus reducing the risk of cancer and cardiovascular diseases [30,31]; improving immunity [32]; and showing neuroprotective, anti-inflammatory and antimicrobial properties [33]. The most common carotenoids of tomato fruits are β-carotene, lutein and lycopene, which are synthesized at the highest levels in red varieties [34]. The data shown in Figure 3 indicate that the β-carotene/lycopene/lutein ratio in ordinary growing conditions of cultivar Podmoskovny reached 1.00:5.02:1.01 in open field and 1.00:7.50:1.03 in greenhouse, with a significantly higher total carotenoid content in the former case (152.7 mg 100 g$^{-1}$ d.w. vs. 127.4 mg 100 g$^{-1}$ d.w. with $p < 0.01$ significance). The corresponding carotenoid ratio in fruits of plants derived from space-stored seeds was equal to 1.00:3.90:1.04 in open field conditions and 1.00:7.10:0.94 in greenhouse. Taking into account that lycopene is a precursor in the biosynthesis of β-carotene and lutein, it may be inferred either that the

reduction of lycopene biosynthesis took place in greenhouse or that the transformation of lycopene to β-carotene and lutein in these conditions was reduced both in control plants and in plants derived from space-stored seeds. Furthermore, the significant increase in total carotenoid content due to the spaceflight effect on tomato seed quality proves the existence of oxidative stress in plants derived from space-stored seeds, which is reflected by the increase in the total carotenoid content by 15% in tomato fruits produced in open field and by 28% in greenhouse (Figure 3).

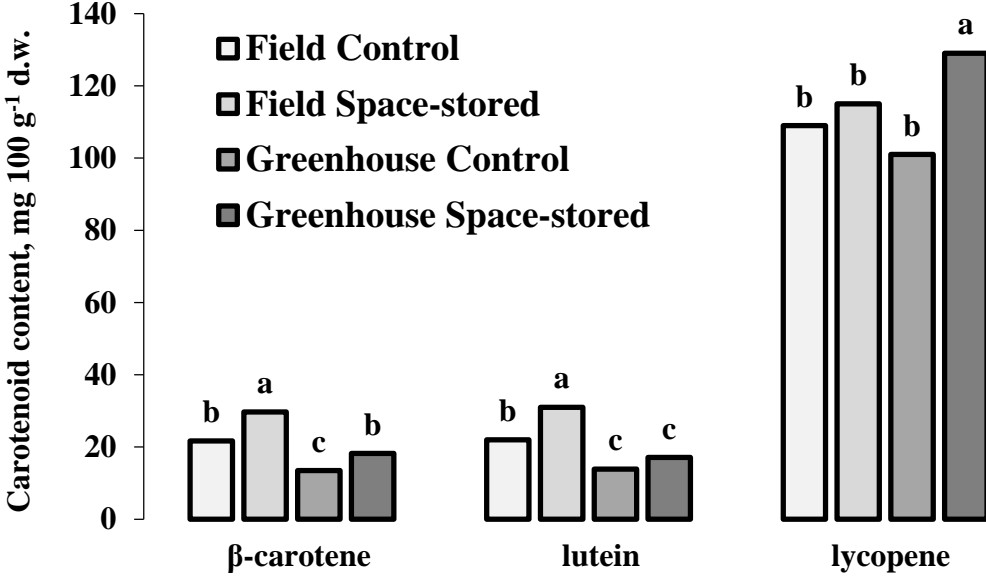

**Figure 3.** Carotenoid profile of tomato fruits obtained from control plants and plants derived from space-stored seeds grown in open field and in greenhouse. For each carotenoid, values with the same letters do not differ statistically according to Duncan test at $p < 0.05$.

Ascorbic acid (AA) plays an important role in plant antioxidant defense, being a key substrate for the detoxification of reactive oxygen species [35]. Overproduction of reactive oxygen species (ROS) in plants under stress conditions is reduced by the production of enzymatic and nonenzymatic antioxidants. In this respect, AA is one of the universal nonenzymatic antioxidants having substantial potential for scavenging ROS and also modulating a number of fundamental functions in plants both under stress and unstressed conditions. The data of ascorbic acid content in control tomato fruits and tomato fruits derived from space-stored seeds (Table 3) are given on a dry weight basis in order to make an adequate comparison between the treatments having different levels of dry matter. In this respect, the results indicate rather small differences in tomato fruit AA content between control plants and plants derived from space-stored seeds, which reached 6.3% in open field and 20% in greenhouse. In the latter conditions, a similar increase in total antioxidant activity was recorded (21.1%), whereas no AOA differences arose in open field.

Phenolics in tomato fruits, represented by chlorogenic acid and quercetin [34], are important antioxidants both for plant integrity and human health [33] and the major contributors to antioxidant activity in tomatoes [36]. In the present investigation, total phenolics are of special interest as the differences in their content between control plants and plants derived from space-stored seeds reached the highest values of 26.7% in greenhouse and 36.3% in open field. Contrary, no differences in phenolic levels were observed between tomato fruits grown in greenhouse and open field. Interestingly, among the antioxidants studied, polyphenols proved to be the most sensitive to long-term consequences of spaceflight.

**Table 3.** Antioxidant compounds and activity of tomato fruits obtained from control and spaceflight-exposed seeds, in greenhouse and in open field.

| Parameter | Control Seeds | | Space-Stored Seeds | |
|---|---|---|---|---|
| | Greenhouse | Field | Greenhouse | Field |
| | Fruits | | | |
| AA (mg 100 g$^{-1}$ d.w.) | $399 \pm 30$ c | $537 \pm 40$ ab | $479 \pm 35$ b | $571 \pm 42$ a |
| AOA (mg GAE g$^{-1}$ d.w.) | $18.0 \pm 1$ b | $22.5 \pm 1$ a | $21.8 \pm 1$ a | $22.5 \pm 1$ a |
| TP (mg GAE g$^{-1}$ d.w.) | $13.4 \pm 1$ b | $13.5 \pm 0.9$ b | $17.1 \pm 1$ a | $18.4 \pm 1$ a |
| | Roots | | | |
| AOA (mg GAE g$^{-1}$ d.w.) | $10.7 \pm 0.8$ a | $12.6 \pm 1.0$ a | $6.5 \pm 0.3$ b | $8.7 \pm 0.4$ c |
| TP (mg GAE g$^{-1}$ d.w.) | $7.0 \pm 0.5$ a | $6.8 \pm 0.4$ a | $6.5 \pm 0.3$ ab | $6.0 \pm 0.3$ b |

AA: ascorbic acid; AOA: antioxidant activity; TP: total phenolics. In each row, the values with the same letters do not differ statistically according to Duncan test at $p < 0.05$.

The comparison between the antioxidant status of tomato fruits and roots revealed the opposite tendency in antioxidant distribution, especially pronounced in AOA values. Indeed, while root TP decreased in plants derived from space-stored seeds, reaching only 13.3% in open field and 7.7% in greenhouse, root AOA values decreased by 44.8% and 64.6%, respectively (Table 3). The latter phenomenon indirectly indicates the possibility of antioxidant redistribution in plants derived from space-stored seeds, with the AOA showing a decrease in roots and increase in fruits.

*3.3. Monosaccharides, Organic Acids, Taste*

Organic acids together with sugars are the main soluble components of ripe fruits and have a major effect on taste, being responsible for sourness and contributing to the flavor; it is well known that sugar and organic acid contents are positively correlated in tomato fruit [37]. Contrary to soluble carbohydrates, which are translocated into the fruits as products of photosynthesis, organic acids are synthesized predominantly in fruits from imported sugars. Fruit acidity, measured by titratable acidity and/or pH, is an important component of fruit organoleptic quality [38], and it is connected to the presence of organic acids, with malic and citric acids being the main acids found in most ripe fruits [39]. Both genetic and environmental variations affect organic acid accumulation in tomato fruits. Notably, nitrate accumulation may stimulate organic acid biosynthesis [40], and in this respect, decreased levels of nitrates and TDS result in decreased TA values.

Though recent investigations indicated a significant role of total antioxidant activity (AOA), total phenolics (TP) and ascorbic acid (AA) content in tomato fruit taste [40,41], only one equation between sugar and organic acids content is presently used for appropriate evaluation [28,34,42]. Monosaccharides (glucose and fructose, present at equimolar ratios) are known to dominate in tomato fruits with a negligible amount of disaccharides [43]. Differences in titratable acidity (TA) between control fruits and fruits derived from space-stored seeds were as much as 35.1% in open field and 50% in greenhouse conditions. Citric acid is known to prevail in tomato fruit organic acids [33], whereas malic and oxalic acids contribute to a lesser extent to the titratable acidity value [44]. As can be inferred from the above data, both TDS and TA demonstrated similar differences between control plants and plants derived from space-stored seeds; i.e., they were higher in greenhouse fruits than in the open field ones. Furthermore, according to taste maturity index (TM = TS:TA) [28], fruit of control plants and plants developed from space-stored seeds did not differ at the stage of maturity, with the TM index being equal to 9.00–9.47 for greenhouse plants and 7.56–7.60 for those grown in open field. In this respect, the taste index of fruit from control plants and plants derived from space-stored seeds, calculated according to Navez et al. [28], revealed higher values in greenhouse and lower in open field (Table 4).

**Table 4.** Monosaccharide and organic acid contents, and taste index (TI) of tomato fruits obtained from control and spaceflight-exposed seeds, in greenhouse and in open field.

| Parameter | Control Seeds | | Space-Stored Seeds | |
|---|---|---|---|---|
| | Greenhouse | Field | Greenhouse | Field |
| Brix (% f.w.) | 7.1 ± 0.4 a | 3.8 ± 0.2 b | 4.2 ± 0.2 b | 2.8 ± 0.1 c |
| TS (% f.w.) | 7.1 ± 0.3 a | 3.8 ± 0.2 c | 4.5 ± 0.3 b | 2.8 ± 0.2 d |
| TA (% f.w.) | 0.75 ± 0.04 a | 0.50 ± 0.03 b | 0.50 ± 0.03 b | 0.37 ± 0.02 c |
| TI | 1.22 | 0.88 | 0.92 | 0.78 |
| TM | 9.47 | 7.60 | 9.00 | 7.57 |

TS: total sugar; TA: titratable acidity; TI: taste index; TM: taste maturity index. In each row, the values with the same letters do not differ statistically according to Duncan test at $p < 0.05$.

The recorded monosaccharide trends were similar to TDS changes (Figure 1), as monosaccharides are one of the main components of water-soluble extracts in tomato fruits. Notably, the results indicate a significant decrease in fruit sugar content of plants derived from space-stored seeds, as much as 35.7% and 57.8% in open field and greenhouse, respectively (Table 4), which is in contradiction with the hypothesis of oxidant stress development. According to literature reports [45], carbohydrates also participate in plant antioxidant defense, and their amount usually increases as a consequence of stress conditions. The opposite differences recorded in the present research for fruits of plants derived from space-stored seeds were statistically significant for values calculated on a dry weight basis, i.e., 12.4% decrease for fruits grown in greenhouse and 22.1% decrease for fruits grown in open field.

### 3.4. Elemental Composition

Changes in fruit elemental composition were revealed only for Fe and Cu content, whose concentrations in fruits derived by space-stored seeds decreased by 1.5 and 1.24 times compared to control plants in greenhouse conditions and by 1.4 and 1.3 times in open field (Table 5). No differences were detected for Mn and Zn content in fruit between control plants and plants derived from space-stored seeds.

**Table 5.** Elemental composition of tomato fruits obtained from control and spaceflight-exposed seeds, in greenhouse and in open field (mg kg$^{-1}$ d.w.).

| Growing Environment | Seed Origin | Zn | Mn | Fe | Cu |
|---|---|---|---|---|---|
| Greenhouse | control | 7.0 ± 0.5 ab | 5.5 ± 0.4 bc | 45.8 ± 3.7 a | 3.1 ± 0.2 a |
| | space-stored | 6.2 ± 0.4 b | 4.8 ± 0.3 c | 29.8 ± 2.0 b | 2.5 ± 0.1 b |
| Open field | control | 8.7 ± 0.7 a | 6.8 ± 0.5 a | 39.8 ± 3.1 a | 3.0 ± 0.2 a |
| | space-stored | 7.8 ± 0.7 a | 6.0 ± 0.5 ab | 28.4 ± 0.2 b | 2.3 ± 0.1 b |

Within each column, the values with the same letters do not differ statistically according to Duncan test at $p < 0.05$.

Iron is an essential micronutrient for almost all living organisms, playing a critical role in DNA synthesis, photosynthesis and respiration [46]. In plants, iron is involved in chlorophyll synthesis, and it is essential for maintaining chloroplast structure and function [47]. As far as Cu is concerned, this microelement is known to serve as an essential cofactor in plant proteins, performing pivotal functions in plant cells by participating in electron transport [48]. Based on the above reports, reduced levels of Cu and Fe may be connected to decreased levels of monosaccharides in tomatoes from space-stored seeds [48].

### 3.5. Relationships between the Analyzed Parameters

From the results of biochemical and mineral analyses, important characteristics of plants grown from space-kept seeds were revealed (Figure 4). The high levels of antioxidant parameters and the decrease in Fe and Cu accumulation are in agreement with the existence of significant oxidative stress. The decrease in sugar, dry matter and organic acid contents

is in accordance with lower taste index values in fruits derived from space-stored seeds. The controversial aspect lies in the fact that despite the reduced accumulation of Fe, Cu, dry matter and carbohydrates, the plants grown from space-kept seeds demonstrated higher yield, fruit weight and plant height.

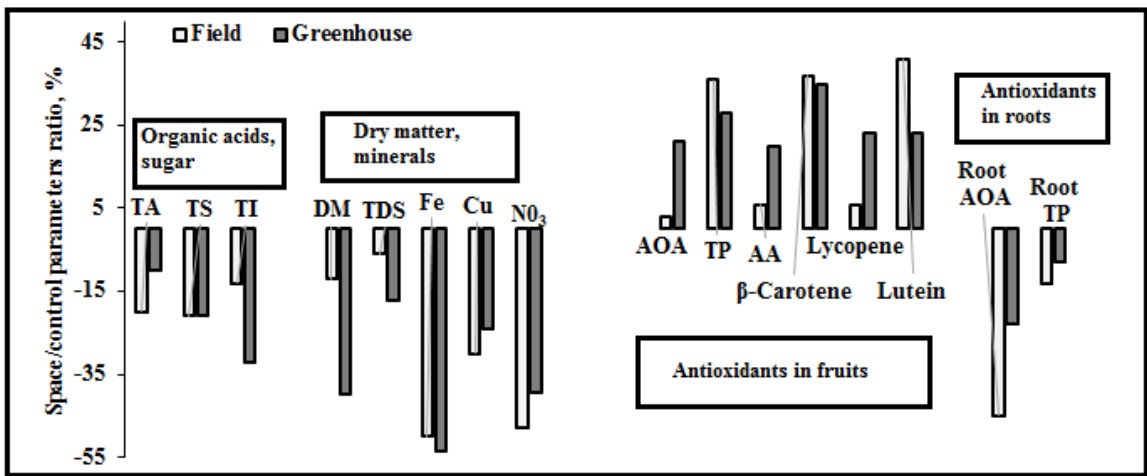

**Figure 4.** Differences in biochemical parameters and mineral content between control plants and plants derived from spaceflight-treated seeds grown in open field and in greenhouse.

On the other hand, the calculation of dry fruit weight indicated extremely small differences, or a lack of difference, between control plants and plants derived from space-stored seeds. In this respect, the fruit weight of control plants and plants grown from space-kept seeds reached $6.67 \pm 0.7$ g and $5.17 \pm 0.5$ g d.w., respectively, in greenhouse and $3.69 \pm 0.3$ g and $3.56 \pm 0.3$ g d.w., respectively, in open field. The latter results indicate the increased dilution in fruits derived from space-stored seeds, as a long-term consequence of the effect of spaceflight on seed quality.

## 4. Conclusions and Future Challenges

Detailed biochemical and mineral characteristics of tomato fruits grown from space-kept seeds revealed the existence of significant oxidative stress in the plants, which was reflected in metabolic antioxidant content changes, a decrease in fruit quality and an increase in fruit yield. The phenomenon relevant to the recorded lowering of biological indicator values resulting from the space-kept seed utilization is dramatically important, though further studies are necessary to evaluate the associated mechanisms and modifications, particularly regarding antioxidant enzyme activity and carbohydrate profile in fruits.

**Author Contributions:** Conceptualization, E.D., G.N. and G.C.; data curation, T.Z.; formal analysis, T.Z. and G.N.; investigation, N.G., M.A., A.K. and I.K.; methodology, N.G., G.C. and E.D.; supervision, A.S. and G.C.; validation, G.C.; draft manuscript writing, N.G., M.A. and E.D.; manuscript revision and final editing, E.D., N.G. and G.C. All authors have read and agreed to the published version of the manuscript.

**Funding:** This research did not receive any grants from public, commercial or not-for-profit agencies.

**Institutional Review Board Statement:** Ethical review and approval were waived for this study, due to the involvement in the authorship of the Istitutions providing the seeds and managing the present research which did not require unusual materials of methods.

**Informed Consent Statement:** Informed consent was obtained from all subjects involved in the study.

**Data Availability Statement:** Not applicable.

**Conflicts of Interest:** The authors declare no conflict of interest.

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
