# Peer review of "Effect of Spaceflight on Tomato Seed Quality and Biochemical Characteristics of Mature Plants"

_horticulturae, doi:10.3390/horticulturae7050089_

Round 1
Reviewer 1 Report
Main comments:
The research undertaken in the manuscript is interesting and many time and labor-intensive measurements have been carried out increase its value.
In the manuscript I noted errors of a technical rather than substantive nature, some methods require more detailed description.
Line 19: “MIR space station” (should be the Mir space station). Mir is a name that does not come from abbreviations of words. Mir is a proper name derived from the word world or peace. Also Lines: 74,77,80.
Line 39: indent a paragraph too long
Line 74: my suggestion to add (grown in the greenhouse and in field)
Line 87; should be- an heated
Line 91 and 113: is temperature (oC) but line 82 and 117 is Ok (°C)
Line 173-174: Measurement provides absorbance. It requires clarification what means: E1% 1 cm and the numbers: (2580; λ = 450 nm), (3470; λ = 474 nm).
Line 155: (Econix, Russia) as line 146
Line 175 (Sigma, country)
Line 179: Statistical Package for the Social Sciences (SPSS) software version 21. (IBM, USA)
Line 187-188: Table 1: In the title of the table 1 should be also provide information that tomato grown from control and spaceflights exposed seeds .
Line 202: In the title of the table 2 should be also provide information that tomato grown in greenhouse and field conditions and plants from control and spaceflights exposed seeds.
Line 212-213 : In the title of the figure 1 should be also provide information that tomato grown from control and space stored seeds. Similar in line 219-220 (Fig.2).
Line 240: should be: by 15% in tomato fruits from open field and by 28% from greenhouse
Line 243: Figure 3: should be: Carotenoids profile of tomato fruits from control and space stored seeds.
Line 267 and line 311: In the title of the table 3 and 4 should be also provide information that tomato grown in greenhouse and field conditions and plants from control and spaceflights exposed seeds.
Line 311: table 4: Sugar TS (% f.w.)
Line 320: In the title of the table 5 should be also information that tomato grown in greenhouse and field conditions.
Line 320: in the first column of Table 5, change to Field instead of Open field
Line 344: In the title of the figure 4 should be also provide information that tomato grown in greenhouse and field conditions
Author Response
Answers to Reviewer 1 comments
Dear Reviewer, we wish to thank you for the useful comments and recommendations contributing to improve our manuscript. All the changes performed inside the text have been marked with red color.
Comments and Suggestions for Authors
Main comments:
The research undertaken in the manuscript is interesting and many time and labor-intensive measurements have been carried out increase its value.
In the manuscript I noted errors of a technical rather than substantive nature, some methods require more detailed description.
- Line 19: “MIR space station” (should be the Mir space station). Mir is a name that does not come from abbreviations of words. Mir is a proper name derived from the word world or peace. Also Lines: 74,77,80.
Answer: the Reviewer comment has been addressed.
- Line 39: indent a paragraph too long
Answer: the aforementioned paragraph has been divided into three paragraphs.
- Line 74: my suggestion to add (grown in the greenhouse and in field)
Answer: addressed.
- Line 87; should be- an heated
Answer: ‘unheated’ is the correct word in our meaning intention.
- Line 91 and 113: is temperature (oC) but line 82 and 117 is Ok (°C)
Answer: addressed.
- Line 173-174: Measurement provides absorbance. It requires clarification what means: E1% 1 cm and the numbers: (2580; λ = 450 nm), (3470; λ = 474 nm).
Answer: specific absorption E1%1 cm means the absorption value for 1% solution of a compound in spectrophotometer cuvette of 1 cm width. As the abbreviation is widely used in analytical chemistry, we just added ‘specific absorption’.
- Line 155: (Econix, Russia) as line 146.
Answer: addressed.
- Line 175 (Sigma, country)
Answer: ‘Japan’ has been added.
- Line 179: Statistical Package for the Social Sciences (SPSS) software version 21. (IBM, USA)
Answer: addressed.
- Line 187-188: Table 1: In the title of the table 1 should be also provide information that tomato grown from control and spaceflights exposed seeds .
Answer: addressed.
- Line 202: In the title of the table 2 should be also provide information that tomato grown in greenhouse and field conditions and plants from control and spaceflights exposed seeds.
Answer: the title has been changed as follows: ‘. Plant height, Fruit yield and dry weight, disease occurrence of tomato plants grown in open field and in greenhouse from control and spaceflights exposed seeds’.
- Line 212-213 : In the title of the figure 1 should be also provide information that tomato grown from control and space stored seeds. Similar in line 219-220 (Fig.2).
Answer: addressed.
- Line 240: should be: by 15% in tomato fruits from open field and by 28% from greenhouse
Answer: addressed.
- Line 243: Figure 3: should be: Carotenoids profile of tomato fruits from control and space stored seeds.
Answer: addressed.
- Line 267 and line 311: In the title of the table 3 and 4 should be also provide information that tomato grown in greenhouse and field conditions and plants from control and spaceflights exposed seeds.
Answer: addressed.
- Line 311: table 4: Sugar TS (% f.w.)
Answer: addressed.
- Line 320: In the title of the table 5 should be also information that tomato grown in greenhouse and field conditions.
Answer: addressed.
- Line 320: in the first column of Table 5, change to Field instead of Open field
Answer: addressed.
- Line 344: In the title of the figure 4 should be also provide information that tomato grown in greenhouse and field conditions
Answer: addressed.
Reviewer 2 Report
The paper of Dzhos et al. concerns the evaluation of the quality of adult tomato plants grown from seeds kept for six moths in a MIR space station. The authors present the results of several biochemical analyzes, such as the content of ascorbic acid, polyphenols, TDS, nitrates, monosaccharides, and measurements of plant dry weight, antioxidant activity, etc.
In my opinion, it is an interesting, well-written work that is part of the research on the influence of space radiation and microgravity on the condition of plants. The obtained results are a valuable extension and supplement to information on the impact of relatively short-term storage of seeds in space on plant development.
I only have a few minor comments and doubts:
- What was the control in the conducted research? In Materials and Methods, the authors describe very precisely the origin of space-treated seeds. But unfortunately, they do not explain what seeds were used as control material for the research.
- Table 1 shows the phenological development of control plants and plants grown from space-stored seeds. Would it not be better to present in a table instead of dates the number of days of cultivation counted from germination or sowing of seeds?
- Table 2 and the text (lines 193-195) talk about plant disease tolerance. How was this assessed? there is no adequate explanation in the methods.
- The authors conclude that “plants grown from space-kept seeds revealed the existence of significant oxidative stress”. Is this conclusion based solely on the evaluation of antioxidant activity and the measurement of phenolic compounds and ascorbic acid? Maybe it would be good to measure the level of free radicals and/or the activity of antioxidant enzymes (peroxidases, dismutase, etc.)?
Author Response
Answers to Reviewer 2 comments.
Dear Reviewer, we wish to thank you for the useful comments and recommendations contributing to improve our manuscript. All the changes performed inside the text have been marked with red color.
Comments and Suggestions for Authors
The paper of Dzhos et al. concerns the evaluation of the quality of adult tomato plants grown from seeds kept for six moths in a MIR space station. The authors present the results of several biochemical analyzes, such as the content of ascorbic acid, polyphenols, TDS, nitrates, monosaccharides, and measurements of plant dry weight, antioxidant activity, etc.
In my opinion, it is an interesting, well-written work that is part of the research on the influence of space radiation and microgravity on the condition of plants. The obtained results are a valuable extension and supplement to information on the impact of relatively short-term storage of seeds in space on plant development.
I only have a few minor comments and doubts:
- What was the control in the conducted research? In Materials and Methods, the authors describe very precisely the origin of space-treated seeds. But unfortunately, they do not explain what seeds were used as control material for the research.
Answer: The following sentence ‘Control seeds of cultivar Podmoskovny were kept at room temperature in the laboratory of the Emanuel Institute of Biochemical Physics since 1998’ has been added to Material and Methods section.
- Table 1 shows the phenological development of control plants and plants grown from space-stored seeds. Would it not be better to present in a table instead of dates the number of days of cultivation counted from germination or sowing of seeds?
Answer: We have modified Table 1, using the number of days of cultivation counted from seed sowing and deleting the last line in order to exclude repetition.
- Table 2 and the text (lines 193-195) talk about plant disease tolerance. How was this assessed? there is no adequate explanation in the methods.
Answer: We have added the sub-section ‘2.2. Resistance to phytophtorosis’ in Materials and Methods.
- The authors conclude that “plants grown from space-kept seeds revealed the existence of significant oxidative stress”. Is this conclusion based solely on the evaluation of antioxidant activity and the measurement of phenolic compounds and ascorbic acid? Maybe it would be good to measure the level of free radicals and/or the activity of antioxidant enzymes (peroxidases, dismutase, etc.)?
Answer: The antioxidant status of plants was assessed by the total antioxidant activity (AOA), as well as the content of polyphenols, ascorbic acid and carotenoids in tomato fruits. Accordingly, the changes in metabolic antioxidants prove the existence of oxidative stress in plants. However, we are aware that it will be additionally informative to analyze the activity of antioxidant enzymes in future investigations, as we expressed in the Conclusions section.
On the other hand, the partial contradictory results relevant to carbohydrate changes in tomato fruit as a result of spaceflight effect on seed quality suggests the complexity of the topic. Indeed, according to literature reports, carbohydrates also participate in plant antioxidant defense and their amount usually increases as a consequence of stress conditions. The opposite differences recorded in our research regarding the fruits of experimental plants are statistically significant for values calculated per dry weight (12.4% decrease for fruits in greenhouse conditions and 22.1% in open field).
Based on the aforementioned information, we widened the discussion of plant antioxidants status and changes in carbohydrate content, as well as the Conclusion section.